# Mesodermal Derivatives of Pluripotent Stem Cells Route to Scarless Healing

**DOI:** 10.3390/ijms241511945

**Published:** 2023-07-26

**Authors:** Yulia Suzdaltseva, Sergey L. Kiselev

**Affiliations:** Department of Epigenetics, Vavilov Institute of General Genetics of the Russian Academy of Sciences, 119333 Moscow, Russia; sl_kiselev@yahoo.com

**Keywords:** mesenchymal stromal cells, human pluripotent stem cells differentiation, immunomodulation, tissue regeneration

## Abstract

Scar formation during normal tissue regeneration in adults may result in noticeable cosmetic and functional defects and have a significant impact on the quality of life. In contrast, fetal tissues in the mid-gestation period are known to be capable of complete regeneration with the restitution of the initial architecture, organization, and functional activity. Successful treatments that are targeted to minimize scarring can be realized by understanding the cellular and molecular mechanisms of fetal wound regeneration. However, such experiments are limited by the inaccessibility of fetal material for comparable studies. For this reason, the molecular mechanisms of fetal regeneration remain unknown. Mesenchymal stromal cells (MSCs) are central to tissue repair because the molecules they secrete are involved in the regulation of inflammation, angiogenesis, and remodeling of the extracellular matrix. The mesodermal differentiation of human pluripotent stem cells (hPSCs) recapitulates the sequential steps of embryogenesis in vitro and provides the opportunity to generate the isogenic cell models of MSCs corresponding to different stages of human development. Further investigation of the functional activity of cells from stromal differon in a pro-inflammatory microenvironment will procure the molecular tools to better understand the fundamental mechanisms of fetal tissue regeneration. Herein, we review recent advances in the generation of clonal precursors of primitive mesoderm cells and MSCs from hPSCs and discuss critical factors that determine the functional activity of MSCs-like cells in a pro-inflammatory microenvironment in order to identify therapeutic targets for minimizing scarring.

## 1. Introduction

Wound healing is a well-orchestrated process that occurs in three overlapping phases (inflammation, proliferation, and remodeling), the sequential path of which is supported by the spatial and temporal synchronization of cellular and paracrine activity. In adults, the remodeling phase results in scar formation with a partial loss of the structure and functional activity of the tissue [1]. At the same time, the impact of certain factors can interfere with successful healing and lead to the development of chronic inflammation, keloids, and hypertrophic scars [2,3]. Patients with scars suffer from functional complications and psychological problems and represent a great burden on the healthcare system. Although the cellular and molecular mechanisms underlying wound healing and scar formation are well described, and modern medications and certain techniques allow wounds to heal faster with less scarring than before, truly effective anti-scarring treatments still do not exist [4,5]. Conversely, mammalian fetal tissues throughout the first and second trimesters of development are known to heal rapidly without scar formation [6,7]. This phenomenon has been demonstrated experimentally in animal models including lamb [8], rabbit [9], rat [10], and mouse [11]. Furthermore, the experimental model for human fetal skin wounds was described by Lorenz in 1992 [12]. These studies demonstrated that damaged fetal tissues were able to restore their original structure with normal three-dimensional collagen architecture, complete epithelial covering with normal differentiation, and epidermal appendages [9,13]. Several mechanisms have been proposed to be involved in fetal scarless wound healing, including the influence of the sterile uterine environment, the composition of secreted growth factors and extracellular matrix, the specific fibroblast phenotype, and a reduced inflammatory response [14]. Obviously, the complete regeneration of fetal tissues occurs due to the coordinated action of these mechanisms, the regulation of which is disturbed in fibrosis. Since such studies are strongly limited by ethical problems in relation to humans, the physiological, cellular, and molecular mechanisms of fetal scarless regeneration remain poorly understood.

Comparative studies of the molecular mechanisms of tissue regeneration in adults and fetuses could identify a specific molecular or cellular target in the scarring pathway to generate a new drug treatment for scars, which could potentially improve the lives of many patients. Human pluripotent stem cells (hPSCs) provide the possibility to generate isogenic cellular models corresponding to different stages of human development due to their ability to reproduce the processes of human embryogenesis in vitro. Implementation of this approach seems to be the optimal solution for this purpose.

Tissue regeneration is impossible without the contribution of mesenchymal stromal cells (MSCs) since the restoration of the parenchyma occurs on the molecular framework they generate. Furthermore, the immunomodulatory activity of MSCs has been shown to contribute to the reduction of acute and chronic inflammation. MSCs perform a regulatory function at the site of damage due to their ability to dynamically change their phenotypes, functional activity, expression, and secretory profile under external signals from the microenvironment, thus promoting successful healing [15]. Scarless healing of fetal tissues is logically assumed to occur with the contribution of fetal MSCs (mesodermal progenitors), which phenotypically and functionally differ from those in adults. However, the differences in the molecular mechanisms involving soluble factors and receptor and adhesive intercellular communications, which determine the functional activity of mesodermal cells in a pro-inflammatory microenvironment in fetuses and adults, remain virtually unknown.

Below, we discuss the known mechanisms by which adult MSCs promote wound healing with a focus on the inflammatory response, extracellular matrix (ECM) synthesis, and remodeling. This is followed by a review of the physiological functions of MSCs in the regulation of tissue regeneration described in animal research reports and clinical studies. Probable pathways and molecules involved in physiological, cellular, and molecular mechanisms of fetal tissue regeneration that differ from adults will also be highlighted. Finally, recent advances in the generation of isogenic cellular models corresponding to different stages of human development are described, and new pathways to investigate phenotypic and functional features of adult and fetal mesodermal cells to elucidate the mechanisms responsible for scarless healing are proposed.

## 2. MSCs-Dependent Mechanisms Involved in Regulation of Inflammation and Tissue Regeneration in Adults

Adult MSCs are now widely accepted to contribute to tissue repair due to their multiple paracrine and immunomodulatory effects. MSCs perform a regulatory function in all phases of wound healing due to their ability to dynamically change their expression and secretory profile under the influence of signals from the microenvironment. The cytokines, chemokines, growth factors, and extracellular vesicles secreted by MSCs are involved in the regulation of intracellular signaling cascades that stimulate angiogenesis, recruitment of cells to the site of inflammation, and remodeling of the extracellular matrix.

MSCs can be isolated and successfully expanded in vitro from different tissue sources. Cultured MSCs are characterized by their adherence to plastic, the expression of the specific surface markers CD73, CD90, and CD105, and their capacity to differentiate into adipocytes, chondrocytes, and osteoblasts [16].

During the early inflammatory response, the secretion of pro-inflammatory cytokines interleukin-1 (IL-1), IL-6, and IL-8 is upregulated in MSCs through Toll-like receptor (TLR) activation. However, TLR stimulation may polarize MSCs into differently acting phenotypes. TLR4-primed MSCs are characterized by the expression of transforming growth factor beta (TGF-beta), indoleamine 2,3-dioxygenase (IDO), prostaglandin E2 (PGE2), and collagen deposition, while TLR3 activation leads to elevated secretion of chemokine (C-C motif) ligand 10 (CCL10), CCL5 (RANTES), IL-10, and fibronectin [17]. MSCs have also been shown to exhibit immunomodulatory effects when interacting with immune cells [18,19]. MSCs possess the ability to inhibit T cell proliferation in mixed lymphocyte reactions [20]. They also suppress the differentiation and maturation of dendritic cells (DCs) and promote their polarization towards the anti-inflammatory M2 phenotype [21]. MSCs mediate the induction of a regulatory phenotype in conventional T cells (Treg) [22,23]. We and others have demonstrated that MSCs can induce functional changes in immune cells and thus have the ability to regulate the balance between pro-inflammatory and anti-inflammatory factors produced, generating a cytokine microenvironment necessary for successful healing [24,25,26]. This immunomodulatory activity of MSCs controls the course of inflammation and its transition to the subsequent proliferative phase of tissue repair. The granulation tissue consisting of a provisional extracellular matrix is characterized by intense angiogenesis. MSCs secrete proangiogenic factors (hypoxia-inducible factor (HIF), vascular endothelial growth factor (VEGF), angiopoietin, monocyte chemoattractant protein (MCP), hepatocyte growth factor (HGF), etc.), which promote migration and proliferation of endothelial cells [27,28,29]. MSCs also support the stabilization and maturation of newly formed vessels [30,31,32]. Extracellular matrix proteins such as fibronectin, collagen, glycosaminoglycans, and proteoglycans synthesized by MSCs are also involved in the process of granulation tissue formation [33,34,35]. During the remodeling phase, metalloproteinases secreted by MSCs regulate the reorganization of collagen fibrils [36,37,38]. MSCs can also stimulate the migration and proliferation of keratinocytes due to the secretion of keratinocyte growth factor (KGF) [39]. The differentiation of MSCs into myofibroblasts contributes to wound contraction [40,41].

## 3. Effectiveness of Adult MSCs in Tissue Repair In Vivo

Functional and phenotypic peculiarities of MSCs have been widely characterized in vitro. However, evidence that these data can be extrapolated to the properties of endogenous MSCs in vivo is still poor and fragmentary.

Nevertheless, Kramann et al. used genetic lineage fate tracing in murine models to demonstrate that, upon injury, tissue-resident Gli1+ MSCs are recruited from the endosteal and perivascular niche to become fibrosis-driving myofibroblasts and substantially contribute to organ fibrosis. Genetic ablation of Gli1+ cells substantially ameliorated bone marrow, kidney, and heart fibrosis [42,43].

Preclinical studies in animal models and clinical trials have also demonstrated the therapeutic applicability of MSCs, confirming their crucial role in tissue repair. MSCs have been shown to promote the healing of full-thickness wounds in mice [44], rats [45,46,47], rabbits [48], and sheep [49]. Reduced synovitis and articular destruction were observed in mice with collagen-induced rheumatoid arthritis after systemic delivery of MSCs in comparison with a control group [50,51]. MSCs were shown to improve left ventricle function, increase vascular density, and decrease scar size, left ventricle stroke volumes, and ejection fractions in a rat model of myocardial infarction [52,53]. MSC treatment effectively improved renal function, resulting in reduced albuminuria, glomerular injury, and renal fibrosis, and promoted wound healing in streptozotocin-induced diabetic mice [54,55].

Currently, the therapeutic potential of human MSCs has been widely investigated in clinical trials. These studies show promising results in MSC treatment of different diseases including immunologic disorders, cardiovascular diseases, diabetes mellitus, chronic wounds, burns, aging frailty, etc. A randomized study conducted by us demonstrated the ability of MSCs to reduce chronic inflammation in long-term nonhealing wounds [56]. Other published clinical trial data demonstrated that the administration of MSCs had beneficial effects resulting in accelerated healing and improved functional capacities and quality of life. The potent mechanisms of MSC-based therapy include stimulation of neovascularization, immunomodulatory activity, reduction of fibrosis, and stimulation of endogenous tissue regeneration acting in concert [15,56,57].

## 4. Molecular and Cellular Mechanisms of Fetal Tissue Regeneration

Physiological, cellular, and molecular mechanisms of fetal tissue regeneration are currently poorly understood because the fetus develops in the isolated environment of the uterus. Nevertheless, several studies of wound repair in animal embryos have identified histological differences between wound healing in fetuses and adults. Healing fetal wounds were shown to contain significantly fewer inflammatory cells compared to adult wounds [14,58]. It was demonstrated that reduced migration of circulating neutrophils into the fetal wound is associated with downregulation of adhesion molecule expression [59,60]. Wulff BC et al., 2012, observed that in mice, dermal mast cells in scarless wounds generated at embryonic day 15 (E15) were fewer in number, less mature, and did not degranulate in response to wounding as effectively as mast cells of fibrotic wounds made at embryonic day 18 (E18) [61]. Other researchers noted that murine fetal wounds had fewer activated macrophages, which appeared and disappeared more rapidly compared to adult wounds. They also found the absence of B cells at the fetal wound site [62].

Inflammatory cells attracted to healing fetal wounds generate a specific microenvironment shown to have a diminished IL-6 and IL-8 response compared to that in adults [63,64]. During fetal scarless wound healing, the ratio of TGF-β3/TGF-β1 isoforms was also shown to be much higher than that in adults [65,66]. In addition, fetal wounds are hypoxic compared to adult wounds and have lower levels of VEGF and reduced vascularity [67]. Therefore, the inflammatory response to injury is significantly reduced in fetal wounds.

ECM composition in fetal and adult wounds undergoing healing also differs. Studies have demonstrated that fetal wounds predominantly regenerate with the deposition of type III collagen, which exhibits a basket-weave network of fine fibrils in a glycosaminoglycan-rich matrix resembling the collagen architecture of uninjured skin. In contrast, after wounding of postnatal skin, the disorganized deposition of type I collagen fibers leads to scarring [68,69,70]. Temporal differences in glycoprotein composition were also found to occur during wound healing in fetuses and adults. Tenascin and fibromodulin are deposited more rapidly in fetal wounds compared to adult wounds [71,72,73]. In contrast, decorin expression is downregulated in fetal wounds and increases rapidly with increasing gestational age [74]. Chondroitin sulfate has a diffuse distribution in the ECM during collagen fibril formation in fetal wounds but is not detected at this stage in adult wounds, except for a very small area immediately below the regenerated basement membrane [71]. Fetal wound fluid was also found to have the prolonged presence of significantly elevated levels of hyaluronic acid, up to 3 weeks, compared with adult wounds where it disappeared within 1 week [75,76]. In addition, a higher ratio of matrix metalloproteinases (MMPs) to their tissue-derived inhibitors (TIMPs) in scarless wounds compared to scarring wounds facilitates the migration of fetal cells and promotes extracellular matrix turnover [77].

The ECM not only represents the tridimensional structural environment of tissues but also may regulate the dynamic interplay between resident and recruited inflammatory cells through the conversion of mechanical stimuli into biochemical signals [78]. MSCs were shown to express various components of ECM, including collagens, glycoproteins, and proteoglycans [33,79,80]. Therefore, changes in MSC populations, phenotype, and functional activity can directly influence wound healing processes.

A differential pattern of gene expression for procollagen 1α1 and 3 was demonstrated in mid- and late-gestational fetal fibroblasts (MSC-like cells) in response to TGF-β1. Mid-gestational fetal cells showed decreased expression of procollagen 1α1, while late-gestational cells showed increased procollagen 1α1 and decreased procollagen 3 expression, demonstrating an increased type 3 to 1 collagen ratio in mid-gestational age and upregulation of type I collagen and decreased type III collagen synthesis with advancing gestational age [81,82]. Cell surface discoidin domain receptors-1 and 2 (DDR1 and DDR2) directly bind collagen and may also regulate collagen deposition by transmitting the signals intracellularly. The elucidation of the role of DDRs in scarless fetal wound repair revealed that DDR1 expression levels were downregulated in fetal fibroblasts with increasing gestational age and were inversely correlated with collagen production, in contrast to DDR2, which showed a similar level of expression throughout gestation [81]. 

Integrins mediate adhesive interactions with specific ECM proteins and can trigger signaling pathways that regulate cellular adhesion, motility, contractile capacity, proliferation, and differentiation in response to the rigidity of the substrate [83]. Variations in α-smooth actin and integrin expression patterns were demonstrated between fetal and adult fibroblasts after treatment with different isoforms of TGF-β. The expression of α-smooth actin and α3 and β1 integrin subunits was increased in adult fibroblasts. In contrast, fetal fibroblast cells showed a decrease in α1, α2, and β1 integrin expression but no change in α3 integrin and α-smooth actin expression. Fetal fibroblasts also showed an inhibition of their contractile capacity in comparison with adult fibroblasts [84].

On stimulation with pro-inflammatory cytokines, fetal and adult fibroblasts differentially express the hyaluronan synthase (HAS1-3) responsible for hyaluronic acid production. Exposure to IL-1 and tumor necrosis factor-alpha (TNF-α) induced a marked increase in HAS-1 and HAS-3 transcript levels in adult fibroblasts compared to the relatively muted response of fetal fibroblasts [85]. WNT3a and TGF-β1 treatment resulted in the induction of HAS2 and HAS3 gene expression in fetal fibroblasts and HAS1 and hyaluronidase-2 in postnatal fibroblasts [86]. Fetal fibroblasts were also found to have an approximately four-fold greater density of hyaluronic acid receptors (CD44) than those of adults [87]. 

Some functional differences have also been observed between adult and fetal fibroblasts in the pro-inflammatory microenvironment. After wounding, fetal fibroblasts appear to migrate in wounds and proliferate there much faster than adult fibroblasts [88,89]. A remarkable feature of fetal wounds is the essential absence of myofibroblasts, which contrasts with their emergence in the later fetal and postnatal periods [90,91]. The smooth muscle alpha-actin (α-SMA) levels have been found to be significantly elevated in adult fibroblasts in comparison to fetal fibroblasts. Thus, fetal fibroblasts do not possess myofibroblasts phenotype [92]. Fetal fibroblasts also exhibit a deficient contractile response to the rigid extracellular matrix and transforming growth factor-β1 in comparison with their adult dermal counterparts [84,93]. 

More recently, functionally diverse lineages of fibroblasts were discovered to coexist in the mouse back skin and oral cavity. Embryonic cells that have expressed engrailed 1 (En1) are responsible for the bulk of connective tissue deposition during embryonic development and wound healing and contribute to scarring in various models of wounds. Conversely, En1-lineage-naive fibroblasts were shown to drive dermal development and regeneration, do not participate in scar production, and their numbers decline with age [94,95]. Transplantation of En1-negative fibroblasts into back-skin wounds of adult mice resulted in a more reticular lattice arrangement of ECM and reduced scarring [95]. 

## 5. Isogenic Cellular Models of Different Stages of Human Development

The generation of isogenic cellular models corresponding to different stages of human development will empower opportunities for investigating phenotypic and functional features of adult and fetal mesodermal cells to elucidate the mechanisms responsible for scarless healing. Due to the inaccessibility of human tissues from different stages of embryonic development, the only way to investigate the mechanisms at these stages and recapitulate the ontogenetic processes in vitro is through the use of hPSCs. The promising technology of hPSCs has expanded our strategies in both basic research and clinical applications, enabling the modeling of human development and genetic diseases and the generation of differentiated cells for transplantation into patients. The pluripotent state of embryonic stem cells (Thomson JA et al., 1998) and somatic cells reprogrammed with transcription factors Oct3/4, Sox2, and Klf4, as well as c-Myc (Takahashi K et al., 2007) provides a feasible approach to obtaining cells corresponding to various stages of human development through directed differentiation in vitro [96,97]. Real-time tracking of hPSC differentiation provides a level of experimental availability of different cell models that is unattainable in vivo.

Currently, multiple strategies have been developed for spontaneous and directed differentiation of hPSCs into specialized cells from all three germ layers [98,99,100]. In the case of mesoderm, recent advances have demonstrated the opportunity to derive various types of mesenchymal cell populations from hPSCs. These include MSCs, hematopoietic cells, tenocytes, smooth muscle cells, osteoblasts, and cardiomyocytes [101,102,103,104,105,106,107,108]. Except for hematopoietic cells, MSCs have provided a differentiation potential with the capacity to generate tissues including bone, cartilage, tendon, muscle, adipose tissue, and bone marrow [109]. Although adult MSCs isolated from various tissues possess similar phenotypes and similar proliferative and differentiation potentials, they originate from developmentally diverse cell populations of lineage-specific mesodermal progenitors, which originate from the neural crest, paraxial mesoderm, and lateral plate mesoderm [101,110,111].

Several studies have reported that MSC-like cells can be obtained directly from hPSCs using serum-containing media without any signal control of intermediate stages [112,113,114,115,116,117]. Advanced methods for generating MSC-like cells and mesodermal progenitors follow a linear approach in which hPSCs are differentiated in discrete steps that mimic the sequence of events that occur during development. The identification of both inductive and repressive signal cues that define the sequential steps, through which hPSCs elaborate diversity of mesodermal progeny, facilitates the generation of mesodermal precursors for MSCs. These can be derived through several intermediate stages including primitive streak cell types [118,119], a neural crest lineage [102,120], neuromesodermal progenitors [121], lateral plate mesoderm cells [119,122], paraxial mesoderm, and somites [105].

Mesoderm development in vivo occurs during early gastrulation when the pluripotent cells of the epiblast begin to differentiate into a primitive streak, which then divides into paraxial and lateral mesoderm. Numerous experimental studies in vertebrate embryo models have yielded important insights into the molecules required for triggering mesoderm formation, maintaining mesoderm state, and mesoderm patterning [123,124,125]. The efforts to differentiate hPSCs into various mesoderm cell types in vitro have demonstrated that signaling molecules WNT, fibroblast growth factor (FGF), bone morphogenetic protein (BMP), and ACTIVIN/NODAL are involved in the induction of mesoderm differentiation and alternative mesodermal path navigation of hPSCs [101,126,127,128,129,130]. Primitive streak formation from hPSCs was found to be initiated following the activation of TGFβ, WNT, and FGF signaling. Co-expression of both BRACHYURY and MIXL1 was observed in these primitive streak-like cells [118,128]. Bifurcation of the primitive streak into the lateral and paraxial mesoderm can be induced by countervailing BMP and WNT signals. WNT activation and blocking BMP signaling were shown to abrogate lateral mesoderm and expand paraxial mesoderm. In contrast, WNT inhibition and exogenous BMP induced the lateral mesoderm. Lateral mesoderm cells were shown to express HAND1 and FOXF1. Paraxial mesoderm cells were defined by the expression of DLL3 and MSGN1 [105,122,128,131]. 

The paraxial mesoderm in embryos is segmented into somitomeres. The inhibition of FGF/ERK and WNT drive paraxial mesoderm cells derived from hPSCs toward early somite precursors expressing MEOX1 and FOXC2. Additionally, a homeodomain-containing transcriptional cofactor HOPX was identified as a somite segmentation marker, which was specifically expressed in a subset of somitomere cells but in neither paraxial mesoderm nor early somites [105,128,129]. 

In vivo, early somites are patterned to generate sclerotome and dermomyotome. Specification of hPSCs-derived somitomere cells in vitro was found to be regulated by the cross-antagonized effects of Hedgehog (Hh) and WNT. Hh activation together with WNT inhibition induced a sclerotome cell population that expressed markers PAX1, PAX9, NKX3.2/BAPX1, FOXC2, SOX9, and TWIST1. Conversely, WNT activation together with Hh blockade exclusively specified dermomyotome from somitomere cells [105,125,128].

Neuromesodermal progenitors in the posterior region of the embryo have the bipotential ability to differentiate into both ectodermal and mesodermal cell types and give rise to the paraxial mesoderm. Several studies have demonstrated that neuromesodermal progenitors can be generated from hPSCs by the activation of TGFβ, WNT, and FGF signal pathways. These cells transiently co-express mesodermal and neural crest markers Brachyury, Sox2, SOX9, SOX10, and Hox genes and can differentiate into neurons, melanocytes, MSCs, and osteogenic progenitors [120,121,132].

Moreover, in addition to the above mesodermal precursors existing transiently in these in vitro culture systems, cloned mesenchymal progenitors have been described. Using multifactorial high-throughput screening technology to engineer in vitro microenvironments that allow expansion of an hPSC-derived mesodermally restricted progenitor population, Kumar N. et al. described intermediate mesoderm cells expressing the pan-mesodermal markers MESP1, MIXL1, and LHX, which were capable of differentiating into cell types with renal gene expression patterns but failed to differentiate into lateral plate mesodermal lineages, such as blood and cardiac muscle [133].

A novel clonal progenitor for endotheliocytes and MSCs, designated as mesenchymoangioblast, was identified during mesendodermal differentiation of hPSCs in the semisolid medium in the presence of FGF2. The population of mesenchymoangioblast cells formed compact spheroid colonies and was positive for the expression of mesenchymal cell surface markers including platelet-derived growth factor receptor (PDGFR), CD146, CD90, and CD56 but did not express CD31, CD43, and CD73. Mesenchymal cells from mesenchymoangioblast colonies were able to differentiate into different types of mesenchymal lineages, including MSCs, pericytes, and smooth muscle cells but not hematopoietic cells and cardiomyocytes. Mesenchymoangioblast was shown to be a transient cell population arising from the apelin receptor (APLNR)+ mesodermal cells at the primitive streak stage. Identification of the lineage tree of mesenchymoangioblast-derived mesenchymal cells demonstrated that APLNR+ cells derived from hPSC express the FOXF1, IRX3, BMP4, WNT5A, HAND1, and HAND2 genes, representative of lateral plate/extraembryonic mesoderm but not the markers of paraxial/myogenic (MEOX1, TCF15, PAX3, PAX7) and intermediate (PAX2, PAX8) mesoderm in the embryo [134,135].

Another group utilized the hemangioblast generated from hPSCs as an intermediate cell type in the derivation of a highly potent and replenishable population of MSCs. The comparison of immunophenotype between hemangioblasts and the resulting MSCs highlighted the differences in the expression of surface markers. Most hemangioblast cells were positive in expression for hematopoietic marker CD45 and only part of them was positive for MSC markers CD90, CD105, or CD73, in contrast to MSCs, which were positive for CD90, CD105, or CD73 and negative for CD45 [136].

Following lineage-specific differentiation, hPSC-derived MSCs acquired some epigenetic, phenotypic, and functional features distinct from adult MSCs though they showed a typical MSC surface marker profile and differentiation potential. hPSC-derived MSCs showed higher CD10 and CD24 expression levels than those in adult MSCs [136]. A grained comparison of transcriptomes between hPSC-derived and adult MSCs revealed that developmental transcription factors HOXD1, NKX2-5, LHX2, and FGF12 as well as CD markers leukemia inhibitory factor receptor (LIFR), prostaglandin F2 receptor negative regulator (PTGFRN), and poliovirus receptor (PVR) were specifically upregulated in hPSC-derived MSCs [137]. hPSC-derived MSCs were also found to acquire a rejuvenation-associated gene signature, specifically, the expression of INHBE, DNMT3B, POU5F1P1, CDKN1C, and GCNT2, which are also expressed in hPSCs but not in the parental adult MSCs [138,139]. Epigenetic rejuvenation of MSCs derived from hPSC was confirmed using DNA methylation analysis. DNA methylation profiles of iPSCs and hPSC-derived MSCs were shown to maintain donor-specific characteristics, but tissue-specific, senescence-associated, and age-related DNA methylation patterns were erased during reprogramming. The DNA methylation pattern of hPSC-derived MSCs remained rejuvenated with regard to the DNA methylation pattern of parental MSCs during culture expansion [140]. Several studies indicate that MSCs derived from hPSCs exhibit stronger proliferation than adult MSCs [121,122,136,141]. Depth characterization of hPSC-derived MSCs, by comparing them to adult MSCs using transcriptomics (RNA-seq) and quantitative proteomics, highlighted biological processes relating to their source: cell cycle and nuclear division-related gene pools were enriched in ESC-MSC, consistent with their higher proliferation rate, whereas BM-MSCs showed enhanced extracellular organization related genes [136]. When induced to differentiate into osteoblasts, chondrocytes, and adipocytes in vitro, MSCs derived from hPSC through both lateral and paraxial mesoderm showed significantly improved osteogenic and chondrogenic potential but less adipogenic potential in comparison with adult MSCs [105,119,122]. The HGF expression level was found to be higher in MSCs derived through paraxial mesoderm progenitors compared to lateral mesoderm-derived MSCs, while VEGF-A and bFGF were more strongly expressed in MSCs derived through paraxial mesoderm progenitors compared to lateral mesoderm-derived MSCs [141].

## 6. Immunomodulatory Activity of Mesodermal Stromal Cells

The functional heterogeneity of MSCs can be determined not only by their ontogenetic origin but also by the inflammatory microenvironment. The immunomodulatory and fibrogenic activity of MSCs is mediated by the autocrine and paracrine effects of cytokines and growth factors as well as cell–cell and cell–matrix interactions through several positive and negative feedback loops occurring during the interaction of resident or migrated MSCs with immune cells in the site of acute and chronic inflammation [15,142,143,144]. 

Studies to date have shown that in vitro stimulation of human adult MSCs through TLR receptors induces the expression of genes involved in chemotaxis and inflammatory responses [145,146,147,148]. Other researchers demonstrated that TNF-α is also involved in the functional reprogramming of MSCs towards enhanced angiogenic and osteogenic activity in vitro and in vivo [149,150,151,152]. Alternatively, IFN-γ treatment was found to prime MSC immunosuppressive activity through the induction of the IDO enzyme, which catabolizes tryptophan into kynurenine and promotes the accumulation of tryptophan-derived catabolites contributing to the inhibition of immune cell proliferation [153,154,155,156]. We have shown that direct intercellular contact can also affect the functional status of MSCs [157].

However, the ability of hPSC-derived MSCs to respond to signals from the microenvironment and influence the function of immune cells has not been described in detail for adult MSCs. Nevertheless, several studies have confirmed the close overlapping of secreting immunomodulatory cytokines (IL-6, IL-8, MCP-1, CCL2) and anti-inflammatory mediators (cyclooxygenases-2 (COX-2), IDO, TGF-beta, programmed death-ligand 1 (PDL1)) between adult and hPSC-derived MSCs in a pro-inflammatory microenvironment [121,136,158]. Adult and hPSC-derived MSCs were also shown to possess a similar capacity to inhibit antigen-induced proliferation of peripheral blood mononuclear cells in a mixed culture system and promote the Treg response [121,136,159,160]. 

Several studies have reported that hPSC-derived MSCs possess improved therapeutic efficacy compared to tissue-specific MSCs in different animal models of diseases. hPSC-derived MSCs were demonstrated to outperform bone marrow MSCs in their ability to extravasate and migrate into inflamed tissues and inhibit T cell infiltration accompanied by an increasing Treg ratio in a murine encephalomyelitis model. Intraperitoneal administration of hPSC-derived MSCs resulted in the prevention of neural demyelination and a reduction in clinical symptoms [161]. hPSC-derived MSCs had a superior neuroprotective capacity over amniotic MSCs in a mouse model of brain ischemia. The increased anti-inflammatory potential of hPSC-derived MSCs was attributed to NF-κB-induced IL-13 production [162]. Compared to human adult bone marrow MSCs, transplantation of hPSC-derived MSCs into mice exerted a greater effect on vascular and muscle regeneration and achieved better attenuation of severe hind-limb ischemia [163]. MSCs derived from hPSC through neuromesodermal progenitors displayed much stronger immunomodulatory activity compared to bone marrow MSCs in vivo, as revealed by decreased inflammatory cell infiltration and diminished production of pro-inflammatory cytokines in inflamed tissue in a mouse model of contact hypersensitivity [121]. Moreover, the difference in therapeutic potential between MSCs derived from hPSCs through paraxial and lateral mesoderm progenitors was discovered in mouse models of skin wounds and pressure ulcers. In a mouse model of skin wounds, MSCs derived from hPSCs through lateral mesoderm accelerated wound healing faster than MSCs of paraxial origin. In contrast, treatment with MSCs originated from paraxial mesoderm progenitors more effectively cured pressure ulcers than treatment with lateral-mesoderm-derived MSCs [141]. 

The known mechanisms underlying the immunomodulatory activity of MSCs cannot perfectly describe the complex and multifactorial processes that support successful wound healing. Fibrotic disorders in various tissues are known to be associated with chronic activation of MSCs by pro-inflammatory mediators, which induce their transdifferentiation into myofibroblasts with scar-producing, proliferative, migratory, contractile, immunomodulatory, and phagocytic properties [143]. However, the antifibrotic activity of MSCs is poorly understood. Recently, we reported that equilibrity between inflammatory factors determines human MSC-mediated immunosuppressive effects, and MSC stimulation through different receptor-dependent pathways can induce some synergistic and overlapping functionalities with integrative effects [24]. The important signaling pathways such as Wnt, BMP, TGF-β, FGF, Notch, and Hh, which play critical roles in tissue patterning during embryonic development and maintaining tissue homeostasis in adults, may also be involved in wound healing and fibrosis [164]. The activation and crosstalk of these signaling pathways in MSCs can generate a gene regulatory network that determines cell fate specification, polarization, proliferation, differentiation, migration, and apoptosis. Thus, these signaling pathways may also represent potential therapeutic targets for attenuating MSC fibrotic activity. The generation of isogenic cellular models corresponding to different stages of human development discloses new pathways to investigate phenotypic and functional features of adult and fetal mesodermal cells in pro-inflammatory microenvironments to elucidate the mechanisms responsible for scarless healing.

## 7. Conclusions

Successful or impaired wound healing, as well as the degree of scarring, largely depends on the phenotypic and functional state of MSCs recruited into the wound, which is regulated by pro-inflammatory signals from the external microenvironment. The current knowledge of the antifibrogenic activity of MSCs in fetal wounds does not allow us to identify the mechanisms responsible for scarless healing. We believe that a comparative analysis of functional activity and gene/protein expression in isogenic mesodermal cells of various stages of human development differentiated from iPSCs under receptor-dependent activation of pro-inflammatory signaling pathways will reveal key molecules and therapeutic targets that will minimize scar formation. By elucidating the molecular mechanisms underlying the functional activity of adult and fetal mesodermal cells in pro-inflammatory microenvironments, we can achieve important advances in our understanding of the processes of wound healing and scarring. Continued work in this field will provide new insights into the mechanisms of wound healing and enable the development of novel therapeutics based on precise inhibition or stimulation of elements of the wound healing pathway.

## Data Availability

No new data were created for this review.

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
