# Peer review of "Mesodermal Derivatives of Pluripotent Stem Cells Route to Scarless Healing"

_ijms, 2023, doi:10.3390/ijms241511945_

Round 1
Reviewer 1 Report
1. A number of grammar mistakes were found. It would be better if the authors revised the whole article and polished the language. For example,
Page 7, line 246 “Moreover, in addition to the above mesodermal precursors existing transiently in vitro culture systems, cloned mesenchymal progenitors have been described.” This sentence contains a double “in”.
2. “The smooth muscle alpha-actin(α-SMA) levels have been found to be significantly elevated in adult fibroblasts in comparison to fetal fibroblasts. The specific regulation of alpha-SMA expression prevents the ability of fetal fibroblasts to differentiate into myofibroblasts.” How is alpha-actin levels specifically regulated to prevent fetal fibroblasts' ability to differentiate into myofibroblasts? Perhaps it should be explained in more detail.
3. Background descriptions for wound healing can be strengthened by citing 10.1016/j.cej.2023.141852; 10.1016/j.ijbiomac.2023.124622 and what are the advantages of the current work compared to published articles?
4. “Compared to adult bone marrow MSCs, transplantation of iPSC-derived MSCs into mice exerted a greater effect on vascular and muscle regeneration and achieved better attenuation of severe hind-limb ischemia.” This comparison is not supported by data, especially experiments on human bone marrow mesenchymal stem cells.
Reviewer 2 Report
Congratulations for summarizing multiple advances pertaining to scarless wound healing. Since you are referring to adult and fetal pathways of healing and regeneration, in opinion you should elaborate more extensively on such key signaling pathway as WNT. WNT is a major regulator of organ development and some notes on interference with active WNT signaling impacting regeneration and/or fibrosis in vivo, WNT/beta catenin dependent and independent pathways contributing to cellular phenotypes triggering and facilitating fibrosis (as opposed to scarless healing) deserve more comments. Second, it would be probably welcomed by readers if authors would share their thoughts on how Hedgehog and WNT signaling represent potential therapeutic targets.
